# Office Hours: A Multiday Office Cubicle Dataset for Associative Embodied VQA

## Abstract

Associating objects with their owners and tracking changes over time are essential capabilities for autonomous robots operating in cluttered, visually redundant, and dynamic environments. Yet existing benchmarks focus on static, uncluttered, and synthetic scenes that fail to capture real-world challenges such as inter-workspace ambiguity and subtle intra-workspace changes. To fill this gap, we introduce the *Office Hours* benchmark dataset: a large-scale, two-part video benchmark comprising six robot-filmed walkthroughs of 23 cubicles over five temporal episodes (*global* subset) and handheld recordings of 10 cubicles across 20 temporal episodes (*local* subset). We annotate ∼1,500 object-level changes across four categories (Object Detection, Count, Localization, State Detection) and provide over 1,600 multiple-choice visual question answering (VQA) questions spanning five complementary tasks: Spatial Association VQA, Static Association–Semantic Mapping VQA, Temporal Association VQA, Single-Cubicle-Multi-Temporal VQA, and Multi-Cubicle-Multi-Temporal VQA.

Using Gemini 2.5 Pro as a strong baseline, our experiments reveal persistent shortcomings: on Multi-Cubicle-Multi-Temporal VQA, the accuracy of localization barely exceeds the random guessing level (∼25%), on Single-Cubicle-Multi-Temporal VQA, overall accuracy reaches 56.8%, with object counting and object state change questions remaining challenging; These results, among others, highlight critical gaps in current VLMs' ability in maintaining consistent object associations across space and time.

## 1 Introduction

The ability to identify and localize objects based on natural language descriptions is fundamental for autonomous robots to interact effectively with both their environment and human users. A core challenge in this process is object association—the ability to maintain consistent references to the same object in different spatial and temporal contexts. Consider a surveillance robot monitoring an open office space (Fig. 1). Its task is to track objects distributed across multiple cubicles. A user might ask, "How many monitors are on Daniel's desk?" or "Is Jerry's laptop still in his cubicle?" queries that require the robot to correctly associate named entities with their corresponding physical spaces and belongings. Successfully answering such questions demands not only visual recognition but also an understanding of spatial layout and entity grounding across time.

Despite its importance, most existing datasets [9, 6, 12] for robotic scene understanding focus on static, uncluttered environments. In such settings, object associations are often straightforward, as the clean layout and low visual redundancy reduce ambiguity in both object identity and location. In contrast, real-world office environments—particularly open-plan cubicle farms—pose significantly greater challenges. These environments are densely populated with visually similar cubicles, each

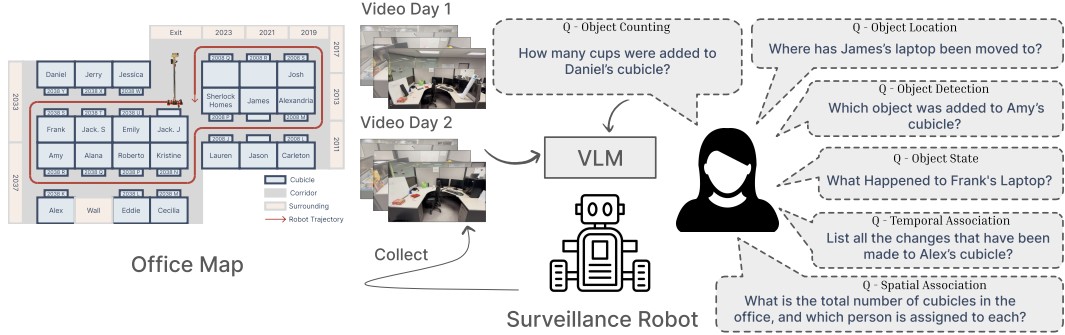

Figure 1: **Overview.** Motivating use case of surveillance EQA, and example questions regarding different types of changes in Office Hours benchmark dataset.

filled with a mix of standard office items and personal belongings arranged in unpredictable ways. Surveillance robots operating in these spaces must distinguish object ownership and track items across both time and space. Addressing this requires solving two interrelated challenges: **spatial association** and **temporal association**.

The **spatial association** problem arises when visually redundant content coexists in a single frame or video sequence. Unlike the curated views of static benchmarks, robots in the real world perceive their surroundings through continuous video streams, often capturing multiple cubicles simultaneously in one frame. This introduces ambiguity when distinguishing which object belongs to which individual or cubicle. To perform robustly in these environments, robots must accurately infer spatial boundaries, associate objects with individuals using both visual and contextual cues, and maintain these associations even when explicit indicators, such as name on whiteboards, are intermittently occluded or only present in other frames. Leveraging Vision-Language Models (VLMs) pretrained on internet-scale datasets have become the leading paradigm to scene understanding and embodied question answering [3]. However, our experiments show that State of The Art (SoTA) VLMs struggle with such spatial association tasks. For example, as illustrated in "Video Day 1" of Fig. 2 , when a cubicle is labeled "Daniel" and the model is asked, "How many monitors does Jerry have?"—despite "Jerry" not appearing in the frame—the correct answer should be "Unable to answer." Yet, both GPT-4o (05122025) and GPT-o3 (05122025) return the number of monitors visible in Daniel's cubicle, incorrectly attributing them to Jerry. Gemini 2.5 Pro Preview (05062025) performs even worse, including a monitor from an adjacent cubicle and counting an iPad. These results reveal a key limitation: current VLMs fail to respect spatial boundaries and struggle to associate named entities with their corresponding physical spaces and belongings.

The **temporal association** problem emerges when models attempt to link objects across different time steps, which often involve changes in camera viewpoint, lighting, and settings (e.g., landscape vs. portrait orientation). VLMs are particularly vulnerable to inconsistencies introduced by these variations. We identify three recurring failure modes: (1) tracking failure due to object misclassification, (2) incorrect associations caused by multiple instances of the same object, and (3) positional misalignment or object disappearance induced by slight changes in camera perspective.

For instance, in "Video Day 2" of Fig. 2, a pile of cables at the cubicle's left corner is misclassified as headsets or game controllers by GPT-o4-mini-high and Gemini 2.5 Pro. We hypothesize that low-confidence predictions vary between frames, leading to false temporal change detection. Another example shown is, when the cubicle's keyboard count increases from one to two (when the original keyboard is removed and two new ones are added), yet the model mistakenly treats this as the original keyboard having simply been moved to a different position and another keyboard being added. Similarly, subtle changes in viewpoint can create the illusion of positional shifts.

A cup visible in an initial wide shot ("Video Day 1") is no longer present in a closer follow-up view ("Video Day 2"). Without robust spatial grounding, the model incorrectly infers that the cup was removed. A model with better spatial-temporal reasoning would recognize that the cup belongs to a neighboring cubicle and should be excluded from the current frame's interpretation.

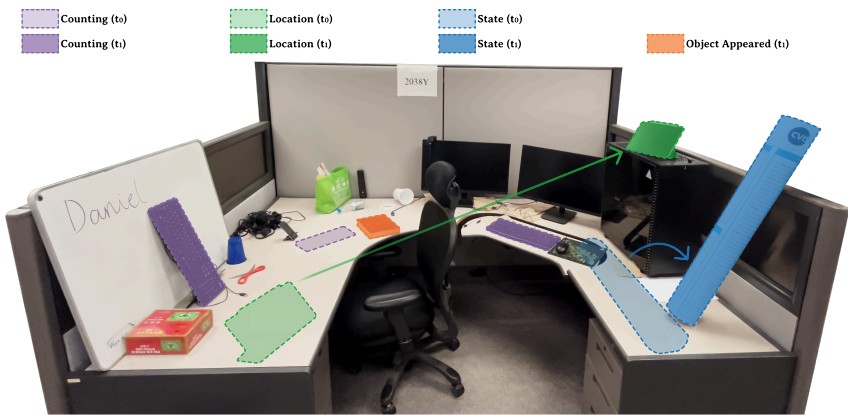

Figure 2: Illustration of the changes in Daniel's cubicle in the dataset. Four primary object-change categories are shown: **Counting Change** (*purple; keyboard quantity increases from one to two*), **Location Change** (*green; the iPad moved from the left desk to atop the black computer case*), **State Change** (*blue; the poster shifts from lying flat to standing upright*), and **Object Appearance** (*orange; a book newly appears*). Light colors indicate initial states at timestamp $t_0$, while darker colors highlight updated states at $t_1$, emphasizing temporal changes.

To address these gaps, we introduce the **Office Hours** benchmark dataset: a large-scale real-world dataset designed to evaluate VLM performance in complex, dynamic office environments from a robotic perspective. Office Hours contains $\sim 1,500$ scene changes in total, including $\sim 500$ changes across 23 cubicles captured over 5 episodes using a robot, and $\sim 1,000$ changes across 10 cubicles collected manually over 20 episodes. Figure Fig. 2 provides an illustrative example of the changes. The dataset is accompanied by more than 1600 Visual Question Answering (VQA) questions assessing fundamental scene understanding, and evaluating a model's ability to understand spatial and temporal association. Through extensive experiments, we primarily study how VLM performs in temporal and spatial association problems.

## 2   Related Work

The emergence of multimodal VLMs, such as RT-2 [1], has significantly advanced robotics by enabling agents to interpret visual scenes and reason about tasks in a generalizable manner. Unlike traditional rule-based systems, which struggle with out-of-distribution scenarios, VLMs [8] leverage joint vision-language representations to perform zero-shot inference across diverse tasks.

Recent robotics-specific VLMs have rapidly evolved to support more complex behaviors. For example, Open X-Embodiment dataset [7] aggregates robot interaction data across varied embodiments, enabling the training of vision-language-action (VLA) models that support cross-embodiment generalization. More recently, Physical Intelligence introduced Pi-0.5 [5], which integrates a VLM with an action expert model to perform long-horizon manipulation in real-world homes. In parallel, navigation-centric VLMs, such as NaVILA [2], incorporate spatial reasoning into language-conditioned navigation, allowing agents to follow high-level instructions in complex, real-world environments.

VQA is a long-standing benchmark for multimodal reasoning and is highly relevant to embodied scene understanding. EmbodiedQA [3] introduced a synthetic household dataset to benchmark spatial and attribute reasoning in closed environments. RoboVQA [11] captured long-horizon video-text demonstrations from humans and robots, focusing on manipulation tasks. HM-EQA [10], built on the Habitat-Matterport 3D (HM3D) dataset [9], improves realism through photorealistic indoor scenes. However, its environments remain overly clean and structured, lacking the clutter, occlusion, and redundancy commonly seen in real-world offices.

To address more complex semantic queries, S-EQA [4] introduced questions involving multiple object states (e.g., "Is the kitchen ready for meal preparation?"), while OpenEQA [6] provides 1,600 human-authored questions covering seven reasoning tasks such as spatial understanding, world

knowledge, and object localization. Nonetheless, most of these benchmarks remain static, feature sparse environments with minimal redundancy, and primarily target household settings that have ready-for-sale cleanness.

The most closely related work to ours is IRef-VLA [12], which explores referential grounding in 3D scenes, including scenarios with ambiguous or imperfect language queries. In contrast, our work focuses on 2D settings, which align more naturally with image-based VLA models trained on large-scale visual datasets. Furthermore, the absence of depth cues, variations in camera viewpoints, and inconsistencies in image quality introduce unique challenges for achieving accurate referential disambiguation in 2D.

# 3 Office Hours: Data Curation

We designed our data collection process to reflect the dynamic nature of real-world office environments. Our goal is to enable robots to better understand scenes over time and perform everyday tasks-such as security checks, item retrieval, and deliveries-that require associating names or cubicles with objects across multiple time instances. We leverage these structured changes not only to capture realistic office dynamics, but also to systematically generate targeted questions that probe a VLM's ability to track and reason about object persistence, movement, and identity across both time and space.

We construct the **Office Hours: A Multiday Office Cubicle Dataset for Associative Embodied VQA**. This benchmark is split into complementary *global* and *local* subsets that *share the same four categories of object-level changes* listed in Table 2.

**Global Changes (inter-cubicle).** We recorded **six** panoramic walk-through videos that each capture all **23** cubicles. Consecutive pairs of videos form **five** temporal episodes (episode $e =$ (video $e-1$, video $e$)). Between episodes we applied object-level manipulations in the physical world in each of the four categories—Presence/Detection, Count, Location, and State/Condition. For instance, a laptop might appear in another cubicle, a set of pens could decrease from five to three, or a monitor could switch from *off* to *on*. Each change is recorded in a category-specific CSV file, and we use VLMs to automatically convert every entry into a multiple-choice question with four answer options plus a "none of the above" choice.

We also introduce **Static Association-Semantic Mapping** questions, which target the VLM's ability to resolve spatial ambiguities in a single video frame. The questions are generated from keyframes extracted from **1** global office video where multiple cubicles are visible and uses a semantic mapping that annotates the robot's current location, visible cubicles from the robot's location, and static landmarks (e.g., large whiteboards, room door numbers). This map is also used to prepend spatial prefixes (cubicle names, e.g. *"From Amy's cubicle..."*) to questions to provide frame-specific spatial context, testing whether VLMs can correctly associate objects with the appropriate cubicles in cluttered scenes.

**Local Changes (inter-cubicle).** For fine-grained temporal reasoning we filmed **10** individual cubicles, capturing **21** short clips per cubicle and therefore **20** temporal episodes each. Here, the same four change categories are applied *within* a single cubicle: objects can newly appear or disappear, their counts can rise or fall, they can be moved to a different spot on the desk, or their state can change (e.g., a laptop lid opens). Each cubicle thus has four CSV logs—again one per category—yielding **40** files in total, and each logged change is turned into a QA pair identical in format to the global subset.

Table 1 summarizes the dataset scale, and Table 2 provides precise definitions of the four change categories for both subsets.

## 3.1 Collecting Video Data

**Recording platforms.** We used two complementary capture methods:

- **BracketBot** - an open-source low-cost 3D printed robot–manually operated by a human pilot.

Table 1: Dataset composition and annotations. "Videos" counts raw clips; "Episodes" counts successive video pairs $(v_{e-1}, v_e)$; "CSV logs" counts files, one per change type.

| Subset | Videos | Episodes | CSV logs | Changes Recorded |
|---|---|---|---|---|
| Global | 6 (panoramic) | 5 | 4 | 490 |
| Local | 215 | 20 per cubicle on average | 40 | 992 |

Table 2: Categories of object-level changes captured in the dataset, organized by granularity: global (across cubicles) and local (within a single cubicle).

| Object-Level Change | Global | Local |
|---|---|---|
| *Presence/Detection* | The appearance or disappearance of object in the video | |
| *Count* | changes in object count (including introducing or removing all instances of object e.g., going from zero to multiple items) from a single cubicle | |
| *Location* | changes in the location of identifiable objects across cubicles | changes in the location of identifiable objects with in a cubicles |
| *State/Condition* | changes in object states, such as orientation or condition and location within a cubicle | changes in object states, such as orientation or condition |

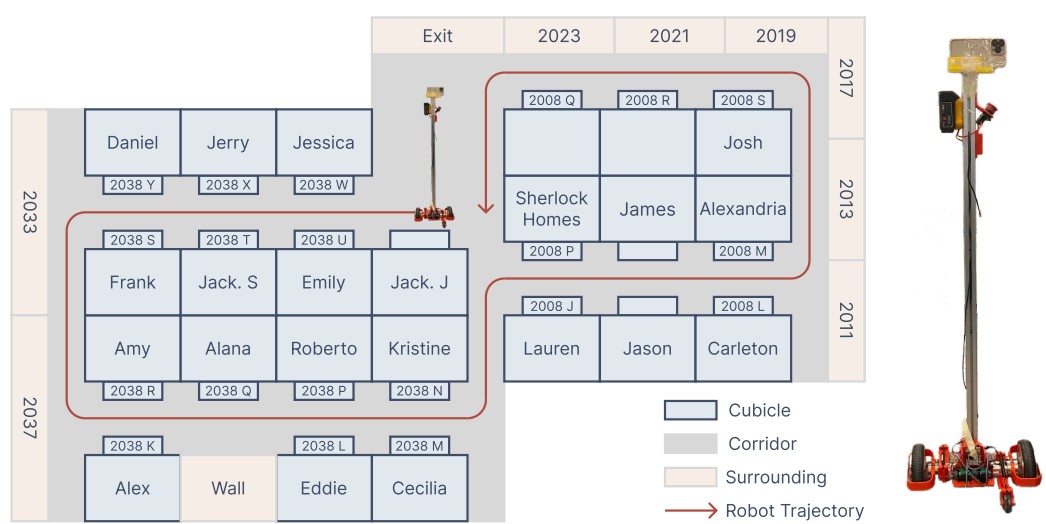

Figure 3: **Left:** Office map with robot video collecting trajectory. **Right:** Bracket Bot used for video collection.

- **Handheld smartphone** – operated by a person, allowing slow, stable pans that fully reveal every surface inside a cubicle.

**Global changes (BracketBot).** The global subset was filmed entirely with BracketBot. Following the route shown in Fig. 3, the robot completes a full loop of the office, recording all 23 cubicles. After every loop we introduced roughly 100 controlled edits—equally divided among the four change categories listed in Table 2—yielding about 500 annotated changes across five temporal episodes. Each change category is stored in its own CSV file (four files total) and later converted into multiple-choice (A–E) questions. Every video is ∼ 10 minutes long, 1080p, and shot with an iPhone 13 Pro Max wide-angle lens.

**Local changes (Handheld).** We chose 10 cubicles and filmed 21 short clips of each, producing 20 temporal episodes. A variety of smartphone models were used to mimic the heterogeneous cameras found on different robots. After each clip we introduced five edits—one per change

category—yielding $\sim 100$ changes per cubicle, evenly distributed across the four categories. Every edit was logged immediately in four per-cubicle Excel sheets and later converted into multiple-choice (A–E) questions identical in format to those of the global subset.

## 3.2 VQA Question Generation

Manually crafting a question for every recorded change is time-consuming. As such we decided to employ a LLM to create the questions. We decided to use Gemini 2.5 Flash (preview 04-17) instead of its ChatGPT o3 due to its larger context window.

**Global Video Changes Questions:** The global changes were partitioned by category into four CSV files: *Object Counting*, *Object Detection*, *Object Location*, and *Object State*. Each CSV and an accompanying prompt were supplied to Gemini, which generated one question per change. A random sample of 20 questions per category was subsequently validated by a human annotator for correctness and clarity.

Every generated question was required to be in a five-option multiple-choice format (A–E) with choice E reading "None of the above" (or equivalent), to demand multimodal reasoning—so that the correct answer could not be inferred from the text description alone—and to hinge on the temporal comparison of two consecutive videos. Examples of the questions created by Gemini are shown in the supplementary material under the section sample questions.

**Static Association-Semantic Mapping Questions:** Focuses on the VLM's ability to associate static visual observations from a single global video with spatial context provided through a semantic map. In addition to the four primary global change categories, we introduce an additional fifth category, *Cubicle/Room Location*, which evaluates spatial reasoning in static scenes. Example of the new category created by Gemini is shown in the supplementary material under the section sample questions.

**Local Video Changes Questions:** For the local changes, an identical pipeline was applied to the change logs of each cubicle, generating a set of fine-grained questions that evaluate object-centric reasoning within confined spatial contexts.

**Office Map Understanding:** Focuses on the VLM's spatial understanding on the global office videos.

**Time Scaled State Question:** Focuses on the VLM's temporal understanding on the local cubicle videos given to the VLM.

## 4 Experiments

Our benchmark is designed to probe how well state-of-the-art VLMs cope with real-world, cluttered office scenes that evolve over time—conditions faced daily by service robots. We evaluate five complementary tasks that together span object recognition, spatio-temporal association, and multi-video reasoning. Note all experiments conducted where done utilizing $300 dollars of free Google Cloud credits.

Our benchmark comprises five complementary tasks:

- **Spatial Association VQA** - question answering over individual episodes requiring the model to count cubicles and associate occupants with their cubicles in cluttered office scenes.

- **Static Association-Semantic Mapping VQA** - question answering using individual keyframes from a global video, grounded with semantic metadata of the robot's location, visible cubicles, and nearby landmarks.

- **Temporal Association VQA** – question answering over pairs of local clips from the same cubicle requiring the model to list changes observed between two videos.

- **Single-Cubicle-Multi-Temporal VQA** – question answering over pairs of local" clips from the same cubicle (intra-cubicle changes).

- **Multi-Cubicle-Multi-Temporal VQA** – question answering over pairs of global" walk-through videos (inter-cubicle changes).

Table 3: Gemini 2.5 Pro (Temperature ($T$=0.0)) answers for cubicle counting and listing tasks for global change videos in which the ground truth number of cubicles is 23.

| Episode | Counting MAPE | Listing Precision | Listing Recall |
|---------|---------------|-------------------|----------------|
| **Average** | 27.5% | 0.491 | 0.394 |

Details of each task are provided in the subsections below.

As a first strong baseline, we benchmark **Gemini 2.5 Pro Preview (05-06-2025)**, currently the top-performing public model on video-understanding leaderboards[1]. Gemini can ingest multiple videos in a single prompt, making it one of the few VLMs capable of handling our episode-pair inputs.

## 4.1 Spatial Association VQA

To quantify Gemini 2.5 Pro's ability to resolve spatial associations in dynamic office scenes, we evaluated its performance on the six global change videos. For each video, the model was prompted to (1) count the number of visible cubicles and (2) list each cubicle's ID alongside its occupant's name. Counting accuracy was measured via Mean Absolute Percentage Error (MAPE) and listing performance was assessed using precision and recall, averaged across episodes.

As shown in Table 3, Gemini 2.5 Pro's counting MAPE is 27.5%, meaning its estimates deviate on average by over a quarter of the true values. On the listing subtask, Gemini 2.5 Pro achieves 0.491 average precision and 0.394 recall—retrieving fewer than half of the true cubicle–name pairs, with limited false positives. These results underscore substantial spatial association challenges: although Gemini can sometimes enumerate and name cubicles correctly, its high error rates and frequent omissions reveal limitations when operating in cluttered, visually repetitive office environments. Results details are included in the supplementary.

## 4.2 Static Association Semantic Mapping VQA

To assess Gemini 2.5 Pro's ability to leverage semantic spatial context for grounded reasoning, we evaluated its quantitative performance on the Static Association-Semantic Mapping VQA dataset using a single global video. Each question was categorized by question type and the model was prompted with the corresponding question, multiple choice options, and the global video (with no image ID association) as context and asked to answer each question accurately using the global video. Similar to Section 4.5, the global video prompted had 720 resolution.

The results are summarized in Table 4. Gemini 2.5 Pro achieves 77.2% overall accuracy, substantially higher than in global-video score. Object Detection (93.0%), Object Location (84.3%), Object State/Attribute (83.3%) are the most accurate, suggesting that the model benefits from stable visual cues. In contrast, Cubical Location (46.7%) and Object Counting (65.5%) were the most challenging to answer, likely due to the difficulty of identifying cubicle boundaries or name tags or cluttered visual scenes.

Table 4: Accuracy (%) of Gemini 2.5 Pro on the *Static Association–Semantic Mapping VQA* task.

| Experiment | Overall | Cubicle Location | Object Detect | Object Location | Object Count | Object State |
|------------|---------|------------------|---------------|-----------------|--------------|--------------|
| Video 0 - 720p | 77.24% | 46.67% | 93.02% | 84.31% | 65.51% | 83.33% |

## 4.3 Temporal Association VQA

To evaluate Gemini 2.5 Pro's capability to resolve temporal associations, we conducted an experiment involving pairs of local videos depicting the same cubicle at different timestamps. For each pair,

---

[1]See the official announcement at https://developers.googleblog.com/en/gemini-2-5-video-understanding/.

Gemini was prompted to identify observed changes and output them in JSON format. Human annotations served as ground truth, capturing actual changes between the video pairs.

We aligned each human-annotated event with the corresponding VLM-generated event by semantically matching object descriptions. Matched events were categorized into three classes: *Matched Change (True Positive)*, *Only in Output (False Positives)*, and *Only in Ground Truth (False Negatives)*. Example of the alignment result can be found in supplementary material in the section temporal change alignment example. Performance was quantified using precision, recall, and F1-score metrics.

Gemini identified a total of 587 correctly matched changes but produced 667 additional incorrect detections and missed 412 genuine changes. This resulted in a precision of 0.47, recall of 0.59, and an overall F1-score of 0.52. These results highlight Gemini's moderate performance in detecting temporal changes, indicating notable limitations in handling object associations accurately over time in dynamic office environments.

### 4.4  Single-Cubicle-Multi-Temporal VQA

For each temporal episode, we provide Gemini with the two walk-through videos of one cubicle $\langle v_{e-1}, v_e \rangle$ and the set of multiple-choice questions derived from the local change logs for that episode and cubicle. Queries follow the structured prompt shown in the supplementary material under the section prompt; we enforce JSON output via Gemini's structured-response schema.

1. **Prompting.** We use the prompt template in supplementary material with temperature $T=0.0$ for deterministic output. If Gemini fails to return valid JSON, we retry with $T=0.25$.

2. **Video preprocessing.** Videos are used at their original recording resolution (1080p) however to reduce the size of them we remove the audio and reduce the frame rate to 10 fps.

3. **Scoring.** Gemini's JSON answer list is compared against ground-truth keys; accuracy is reported per change category and overall.

**Results.** Table 5 reports the aggregate mean across all cubicles. Gemini reaches **56.8 %** overall accuracy—modestly above the global-video score—indicating that even within a single cubicle many changes remain challenging. Object Detection is easiest (63.6 %), followed by Location (61.9 %), State (53.1 %), and finally Counting (48.6 %). These trends align with intuition: estimating exact counts and subtle state changes (e.g., lid-open vs. lid-closed) demand finer spatio-temporal resolution than simply recognizing or localizing an object.

Table 5: Accuracy (%) of Gemini 2.5 Pro on the *Single-Cubicle-Multi-Temporal VQA* task. Asterisks (*) denote runs that required a higher sampling temperature ($T=0.25$) to obtain valid JSON output; all other runs used $T=0.0$

| Cubicle | Total | Object Detection | Object Location | Object Counting | Object State |
|---|---|---|---|---|---|
| **Mean** | 56.8% | 63.6% | 61.9% | 48.6% | 53.1% |
| **Standard Deviation** | 9.4% | 8.1% | 14.1% | 17.2% | 17.4% |

### 4.5  Multi-Cubicle-Multi-Temporal VQA

The *Multi-Cubicle-Multi-Temporal VQA* evaluation mirrors the protocol in Section 4.4: Gemini 2.5 Pro receives the two clips $\langle v_{e-1}, v_e \rangle$ of a temporal episode and must return a JSON list of answers to all multiple-choice questions. The key differences are:

1. **Video preprocessing:** Gemini's free tier limits each file to 100 MB. We therefore transcode both videos to 720p, 10 fps, a Constant Rate Factor of 28, and strip audio. To gauge the impact of resolution, we also run a subset of queries with 1080p videos (10 fps, CRF 20, audio removed) that exceed the 100 MB ceiling on paid accounts.

2. **Map cue variant:** Because 720p footage makes white-board name tags hard to read, we test a second variant in which we append the office-layout map (Fig. 3(a)) alongside the two videos.

3. **Question Batch:** The question batch for an episode contains only one type of change out of *all four* change categories (Detection, Location, Counting, State).

Table 6 shows that Gemini 2.5 Pro struggles most with *Location* questions, scoring just **25.0%**, barely above the *20%* guess rate. Detection and State exceed 50%. Adding an office map yields marginal gains—location improves by 3 points, but overall accuracy drops to 43.8%, suggesting poor use of spatial context. Raising the input resolution to 1080p improves location accuracy to **33.9%** but causes sharp drops in Counting and State performance. It was expected that improving the resolution would not lead to any meaningful improvement because Gemini compresses every video frame to a fixed 258-token representation regardless of resolution[2]. Even under the most favourable setting (720p), the model reaches only 45.2% overall, revealing that current state-of-the-art VLMs still struggle with multi-video reasoning in cluttered, dynamic office scenes.

The Global-VQA task is substantially harder than Local-VQA, with the largest drop in object-location accuracy. This mirrors our spatial-association results (Section 4.1): without reliably identifying cubicles, the model struggles to track objects across workspaces.

Table 6: Accuracy (%) of Gemini 2.5 Pro on the *Multi-Cubicle-Multi-Temporal VQA* task. Asterisks (*) denote runs that required a higher sampling temperature ($T=0.25$) to obtain valid JSON output; all other runs used $T=0.0$.

| Experiment | Overall | Object Detection | Object Location | Object Counting | Object State |
|---|---|---|---|---|---|
| 720p | 45.2% | 54.8% | 25.0% | 40.5% | 61.1 % |
| 720p + Map | 43.8% | 47.0% | 28.2% | 40.5%* | 59.5% |
| 1080p | 36.4% | 47.0% | 33.87% | 22.2% | 43.7% |

# 5   Limitations

While Office Hours provides a challenging benchmark for office-scene reasoning, it remains domain-specific—its focus on cubicle farms may not generalize to industrial, retail, or outdoor settings. The environment is essentially static, with no human actors or dynamic background elements, limiting the dataset's applicability to interactions and real-world lighting changes.

# 6   Conclusion

Real-world robotic applications demand scene understanding that goes beyond static snapshots in controlled settings: robots must navigate cluttered workspaces, recognize both standard and personal items, and maintain object associations across space and time. To address this need, we introduce the "Office Hours" benchmark suite, explicitly designed to stress-test Vision–Language Models (VLMs) on spatial and temporal reasoning in dynamic office environments. We accompany "Office Hours" with a diverse suite of VQA tasks, ranging from change-specific question answering to spatial-association and temporal-tracking experiments.

Evaluating Gemini 2.5 Pro on this benchmark reveals persistent gaps in current VLM capabilities. On inter-cubicle queries, its object-location accuracy hovers just above random chance, indicating severe spatial mislocalization and frequent confusion between neighbouring workspaces. Within single desks, the model struggles with exact counts and subtle state changes, underperforming on both counting and state-change tasks.

These findings underscore critical limitations in the reasoning of today's VLMs—particularly their difficulty in grounding named entities to specific workspaces and in maintaining object identity over time. We believe the "Office Hours" benchmark will provide a valuable resource for systematically quantifying these shortcomings and guiding the development of more robust, embodied scene-understanding models.

---

[2]https://cloud.google.com/vertex-ai/generative-ai/docs/multimodal/video-understanding

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
