# OpenReview forum: "Office Hours: A Multiday Office Cubicle Dataset for Associative Embodied VQA"
_NeurIPS.cc/2025/Datasets_and_Benchmarks_Track — Submitted to NeurIPS 2025 Datasets and Benchmarks Track_

### Official Review · Reviewer_k2TV · 2025-06-21

**Rating:** 5
**Confidence:** 4

**Summary:**

This dissertation considers the ASSOCIATING OBJECTS relationship, synthesizing the challenges of both spatial association and temporal association. A benchmark dataset for Office Hours is introduced. It contains A benchmark dataset of Office Hours is introduced using A dataset of large-scale realistic scenarios is used to evaluate the perceptual performance of VLMs in complex, dynamic office environments. The dataset contains 1600 visual quizzes for fully evaluating different models.

**Dataset Code Accessibility:**

Yes

**Dataset Code Comments:**

https://github.com/Junf137/office_hours
https://www.kaggle.com/datasets/cfe2e8cb6905fe01f377cb55bfdd97d3bd287c3eaa98528681cf3ea3083de9ec

**Ethical Considerations:**

No, there are no or only very minor ethics concerns

**Final Justification:**

The authors have basically addressed my concerns and the problem is of larger value to the study of embodied AI.

**Limitations Weaknesses:**

1. What are the more practical applications of reasoning about object association capabilities in embodied intelligence, such as navigation, skill learning, or human-computer interaction, and can such capabilities enhance these practical applications? Is it possible to design some experiments on some benchmarks or simulation platforms for verification?
2. Are there other scenarios that are encountered in several of the scenarios considered in the paper? For example, living rooms, kitchens, rooms, etc., which are also frequently involved in embodied intelligence applications, are there obvious quantitative, state, and phenotypical changes that this would entail? Is there a significant change in quantity, state, and appearance? What kind of applications can there be for mining their associations?
3. Can the experiment evaluate the effectiveness of some more VLM methods? This would enable the researcher to have a clearer idea of which convenient association the existing work is mainly deficient in exploring? So as to facilitate further analysis of what the causes are.

**Strengths Contributions:**

1. The problems discussed in the paper are relatively novel and have very important research value for scene understanding and perception in dynamic and complex environments.
2. The paper considers the object-change more fully and poses greater challenges
3. The dataset is larger, with better quality, and the experiments in the paper are more adequate.

---

> ### Author Rebuttal · Authors · 2025-07-31
>
> We sincerely appreciate the reviewer’s thoughtful feedback and their engagement with our work. We respond to each point raised below.
>
> ---
>
> ### Concern 1: What practical applications does object association serve?
>
> Thank you for this insightful question. **Object association** is critical in real-world robotics and embodied AI applications, including:
>
> * **Surveillance robots**: Detect missing items or misplaced personal belongings.
> * **Inventory management**: Track item movement across a warehouse or office.
> * **Assistive robotics**: Identify and retrieve objects for users with disabilities.
>
> In the context of surveillance, for example, a robot must not only navigate an office but also understand **object persistence and ownership**, e.g., detecting that *“Jerry’s laptop is missing”* requires both **spatial reasoning** and **memory**. Without object association, even the best navigation capabilities become irrelevant.
>
> While simulation platforms may offer easier benchmarking environments, our goal was to begin with a **real-world setting**. That said, simulation is a compelling future direction, and we are actively exploring the possibility of porting the benchmark to **embodied simulation platforms** such as **Habitat** or **Isaac Sim**.
>
>
> ---
>
> ### Concern 2: Could this extend to other domains like kitchens, living rooms, etc.?
>
> Yes, the core challenges studied in **Office Hours** such as tracking **object identity**, **location**, and **state** over time, also arise in many other real-world settings, including **living rooms**, **kitchens**, **bedrooms**, and **classrooms**, which are widely used in embodied intelligence research.
>
> These environments exhibit:
>
> * **Quantitative changes**:
>   Objects may be added, removed, or counted (e.g., number of dishes, chairs).
>
> * **State changes**:
>   Items may change condition (e.g., lights on/off, laptops open/closed, drawers opened/closed, steak cooked/raw, food spoiled).
>
> **Applications of association mining** in such domains include:
>
> * **Household robotics**:
>   Remembering object locations or tracking usage over time (e.g., has the stove been turned off?).
>
> * **Elder care monitoring**:
>   Associating personal belongings with individuals and monitoring their state or movement for safety.
>
> * **Inventory and organization**:
>   Detecting misplaced or removed items in smart home or workspace environments.
>
> The same **association principles** tested in our benchmark should generalize well to these dynamic domains.
>
>
> ---
>
> ### Concern 3: Could more VLMs be tested to better understand failure cases?
>
> Yes, our benchmark is designed to be **model-agnostic** and **extensible**, and we are actively working to support broader evaluations.
>
> We initially chose to evaluate **Gemini 2.5 Pro** because, to our knowledge, it is one of the only VLMs that **explicitly advertises support for multi-video input** (1–10 videos per prompt) in its product documentation. This makes it well-suited for our benchmark, which involves **multiple cubicle walkthroughs** and **temporal video pairs**.
>
> While models like **GPT-4o**, **LLaVA-Next**, **Video-LLaMA 3**, **Qwen2-VL**, and **InternVL3** are impressive, their documentation did not indicate that they were designed for **multi-video question answering**. We plan to incorporate these and other models in future benchmark updates.

---

### Official Review · Reviewer_99JW · 2025-07-01

**Rating:** 4
**Confidence:** 4

**Summary:**

This paper introduces Office Hours, a large-scale, real-world benchmark for evaluating Vision–Language Models (VLMs) on spatial and temporal object association in cluttered office environments. The dataset has two parts:
1.	Global subset – Six panoramic robot walkthroughs covering 23 cubicles over five temporal episodes, with ~490 annotated object-level changes.
2.	Local subset – Handheld recordings of 10 individual cubicles across 20 episodes, with ~992 annotated changes.
For each change, the authors generate a multiple-choice VQA question, yielding over 1,600 questions spanning five tasks: Spatial Association VQA, Static Association–Semantic Mapping VQA, Temporal Association VQA, Single-Cubicle Multi-Temporal VQA, and Multi-Cubicle Multi-Temporal VQA.

**Dataset Code Accessibility:**

Yes

**Dataset Code Comments:**

Yes, it provided a preview URL (with the verified croissant file) and the code is in GitHub.

**Ethical Considerations:**

No, there are no or only very minor ethics concerns

**Limitations Weaknesses:**

Single Domain: Dataset is limited to an office cubicle farm—lack of diversity may restrict generalization.
No 3D/Depth Data: Only 2D video frames are used; real-world robots often have depth sensors, so the benchmark omits an important cue.
Limited Model Comparison: Only Gemini 2.5 Pro is benchmarked; comparisons with other VLMs or specialized spatio-temporal architectures are missing.
Lack of Error Analysis: No detailed breakdown of failure modes, which would guide future improvements.

**Strengths Contributions:**

Real-World Complexity: First large-scale video dataset in a truly cluttered, multi-workspace office setting, capturing both spatial redundancy and subtle temporal changes.
Comprehensive Tasks: Five carefully designed VQA tasks cover detection, counting, localization, state change, and cross-video reasoning, probing different facets of spatial-temporal understanding.
Thorough Annotation: 1,500 manually logged changes and 1,600 auto-generated, human-validated questions ensure high annotation quality and reproducibility.
Clear Baseline Evaluation: Gemini 2.5 Pro is evaluated across all tasks with appropriate metrics, quantitatively exposing VLM shortcomings.

---

> ### Author Rebuttal · Authors · 2025-07-31
>
> We appreciate the reviewer’s time and insights. Below, we provide a point-by-point rebuttal addressing the key concerns.
>
> ---
>
> ### Concern 1: Domain specificity and lack of dynamic elements
>
> Our benchmark focuses on **spatial-temporal associative reasoning**, which we define as the ability to link entities (e.g., objects and locations) across space and sparse time steps. Office cubicles provide an ideal testbed for this due to several factors:
>
> * **High visual redundancy**
>   Most cubicles look similar, requiring models to rely on sparse cues (e.g., name tags or cubicle codes) to resolve spatial references. This challenges the model’s ability to ground language in cluttered visual environments.
>
> * **Spatial + Temporal Association**
>   Consider the example of tracking *Veronica’s phone* as it is moved to another desk. To answer questions about this change, the model must:
>
>   1. Find Veronica’s cubicle and identify its boundaries (models often mistake objects near a cubicle as belonging inside it),
>   2. Locate the phone within her cubicle,
>   3. Recognize the same phone in a different cubicle in the next video,
>   4. Associate both cubicles with their respective owners.
>
>   This process requires the model to link object identity and ownership across space and time, making it a clear test of spatial-temporal association.
>
> * **Cluttered but structured scenes**
>   Offices contain many overlapping and occluded objects, providing a realistic and challenging environment for evaluating perception under ambiguity.
>
>
> We intentionally limited the benchmark to office space to isolate and evaluate spatial and temporal association capabilities. That being said, office environments are highly relevant to real-world applications such as **surveillance**, **inventory monitoring**, and **workspace assistance**, where tracking object movement and ownership over time is a core requirement.
>
> ---
>
> ### Concern 2: No depth data
>
> The primary purpose of our dataset is to serve as a **benchmark for evaluating Vision-Language Models (VLMs)**, rather than as a training resource, given its relatively limited size.
>
> Since most current VLMs are designed to operate on **RGB inputs**, our dataset focuses exclusively on that modality. We believe that **depth understanding should be inferred by the models themselves**.
>
> One promising direction for future improvement would be to enhance input representations by incorporating **3D information**—for example, by generating **point clouds** from the video data using methods such as **VGGT** or **DUST3R**, and feeding these into the VLM alongside the video frames.
>
> ---
>
> ### Concern 3: Evaluation is limited to Gemini 2.5 Pro. No comparison with open-source VLMs.
>
> Our decision to focus exclusively on **Gemini 2.5** Pro was aligned with the core objective of the Office Hours benchmark: to assess multi-video spatial and temporal reasoning in cluttered real-world scenes.
>
> To the best of our knowledge, **Gemini 2.5 Pro** is one of the only vision-language models (VLMs) that explicitly advertises support for multi-video input (1–10 videos per prompt) in its product documentation. This makes it particularly suited for our benchmark, which involves multiple cubicle walkthroughs and temporal video pairs.
>
> While models like **GPT-4o**, **LLaVA-Next**, **Video-LLaMA 3**, **Qwen2-VL**, and **InternVL3** are impressive, we found no documentation or examples indicating that they were designed for multi-video question answering.
>
> For completeness, we provide a summary below:
>
> | **Model**            | **Specifically Designed for Multi-Video QA?** |
> | -------------------- | --------------------------------------------- |
> | Gemini 2.5 Pro       | ✅ Claims support for 1–10 videos              |
> | GPT-4o               | ❌ Frame-based inference only                  |
> | LLaVA-Next           | ❌ No usage examples with >1 video             |
> | Qwen2-VL-7B-Instruct | ❌ No evidence of multi-video QA usage         |
> | InternVL3            | ❌ No usage examples with >1 video             |
> | Video-LLaMA 3        | ❌ No usage examples with >1 video             |
>
> Given these factors, we chose to benchmark **Gemini 2.5 Pro** first to test whether a model advertised as capable of multi-video understanding could perform spatial and temporal reasoning in cluttered real-world environments. The model’s poor performance on tasks like object localization validates the need for such targeted evaluation.
>
> That said, we fully agree that broader comparisons would strengthen the benchmark, and we are actively working on incorporating other models into the evaluation pipeline in future updates.
>
> ---
>
> ### Concern 4: Lack of error analysis
>
> While we do not provide a detailed failure-mode taxonomy or per-example diagnosis, we believe our paper offers a meaningful level of **quantitative and qualitative error analysis** to understand model limitations.
>
> We report **breakdown scores across four error types**—**Object Detection**, **Counting**, **Location**, and **State**—in three distinct tasks:
>
> * **Static Association–Semantic Mapping VQA** – evaluates spatial understanding in a single global video.
> * **Single-Cubicle-Multi-Temporal VQA** – evaluates fine-grained temporal reasoning within a cubicle.
> * **Multi-Cubicle-Multi-Temporal VQA** – tests joint spatial and temporal reasoning across multiple cubicles.
>
> This decomposition helps isolate which types of association (spatial vs. temporal) and which change categories are most error-prone.
>
> Additionally, we provide focused analysis on one of the most consistent failure cases: **location-based questions** in the multi-cubicle setting. We show that **Gemini struggles not only with tracking**, but also with a prerequisite skill: **reliably identifying cubicle boundaries and correctly assigning ownership**. This is demonstrated by its **high cubicle-counting error** and **low name-association recall** (Section 4.1).
>
> This **root-cause explanation** directly connects basic spatial reasoning failures to the model's observed downstream performance drop.

---

> ### Comment · Reviewer_99JW · 2025-08-09
> **I will maintain my positive score**
>
> Thanks for the author's rebuttal. My concerns have been largely resolved, and I will maintain my positive score.

---

### Official Review · Reviewer_ZLRP · 2025-07-03

**Rating:** 4
**Confidence:** 3

**Summary:**

The submission introduces the "Office Hours" dataset, a large-scale video benchmark for evaluating Vision-Language Models (VLMs) in dynamic office environments.         It comprises robot-filmed global walkthroughs of 23 cubicles over 5 episodes and handheld local recordings of 10 cubicles across 20 episodes, annotating ~1,500 object-level changes.         Accompanied by ~1,600 VQA questions, the dataset tests five tasks: Spatial Association, Static Association–Semantic Mapping, Temporal Association, Single-Cubicle-Multi-Temporal, and Multi-Cubicle-Multi-Temporal VQA.         Experiments using Gemini 2.5 Pro reveal VLMs struggle with cross-space localization (25% accuracy) and temporal state changes, highlighting critical reasoning gaps.

**Dataset Code Accessibility:**

Yes

**Ethical Considerations:**

No, there are no or only very minor ethics concerns

**Final Justification:**

I am still inclined to accept the paper.

**Limitations Weaknesses:**

1. Task Overlap and Design Clarity

Temporal Association VQA and Single-Cubicle-Multi-Temporal VQA both target intra-cubicle changes, but their distinction is unclear (e.g., whether they differ in question format or video pairing).         The paper does not explicitly justify the need for five tasks, leaving readers to infer their complementary roles.

2. Domain and Contextual Constraints

The office cubicle focus limits generalizability to industrial or outdoor scenes.         The lack of human actors or dynamic lighting restricts scenarios requiring interaction-aware reasoning (Sec. 5.0).

3. Baseline and Evaluation Gaps

Experiments only use Gemini 2.5 Pro, missing comparisons with models like GPT-4o or Flamingo.         Additionally, the LLM-generated questions (Gemini 2.5 Flash) may introduce bias, as human validation covers only 20 questions per category (Sec. 3.2).

**Strengths Contributions:**

1.  Real-World Dynamic Scene Modeling

The dataset addresses limitations in static benchmarks (e.g., HM3D, OpenEQA) by capturing cluttered, visually redundant office spaces.         The global subset uses robot-mounted cameras for panoramic cubicle tracking, while the local subset employs handheld devices to simulate diverse viewpoints.         This design reflects real-world challenges like inter-cubicle ambiguity and subtle temporal changes (e.g., object relocations or state shifts, Fig. 2).

2. Five Complementary VQA Tasks

- **Spatial Association VQA**: Tests cubicle counting and occupant mapping in cluttered scenes (Table 3).

3. - **Static Association–Semantic Mapping VQA**: Uses keyframes with spatial metadata (e.g., cubicle names, landmarks) to evaluate grounding (Table 4).
- **Temporal Association VQA**: Requires listing changes between local video pairs (Sec. 4.3).
- **Single-Cubicle-Multi-Temporal VQA**: Focuses on intra-cubicle changes (Table 5).
- **Multi-Cubicle-Multi-Temporal VQA**: Evaluates cross-cubicle tracking (Table 6).
Questions are generated via LLM (Gemini 2.5 Flash) and validated by humans, covering object detection, count, location, and state changes (Sec. 3.2).

4. Reproducible and Open Infrastructure
The dataset provides clear collection protocols (BracketBot robot and handheld devices, Sec. 3.1), question generation workflows, and experimental details (e.g., Gemini’s temperature settings).         Code and data are openly available, enabling replication with $300 Google Cloud credits (Sec. 4.0).

---

> ### Author Rebuttal · Authors · 2025-07-31
>
> We appreciate the reviewer’s time and insights. Below, we provide a point-by-point rebuttal addressing the key concerns.
>
> ---
>
> ### Concern 1: Evaluation is limited to Gemini 2.5 Pro. No comparison with open-source VLMs.
>
> We acknowledge the limitation and appreciate the reviewer’s suggestion. Our decision to focus exclusively on **Gemini 2.5** Pro was aligned with the core objective of the Office Hours benchmark: to assess multi-video spatial and temporal reasoning in cluttered real-world scenes.
>
> To the best of our knowledge, **Gemini 2.5 Pro** is one of the only vision-language models (VLMs) that explicitly advertises support for multi-video input (1–10 videos per prompt) in its product documentation. This makes it particularly suited for our benchmark, which involves multiple cubicle walkthroughs and temporal video pairs.
>
> While models like **GPT-4o**, **LLaVA-Next**, **Video-LLaMA 3**, **Qwen2-VL**, and **InternVL3** are impressive, we found no documentation or examples indicating that they were designed for multi-video question answering.
>
> For completeness, we provide a summary below:
>
> | **Model**            | **Specifically Designed for Multi-Video QA?** |
> | -------------------- | --------------------------------------------- |
> | Gemini 2.5 Pro       | ✅ Claims support for 1–10 videos              |
> | GPT-4o               | ❌ Frame-based inference only                  |
> | LLaVA-Next           | ❌ No usage examples with >1 video             |
> | Qwen2-VL-7B-Instruct | ❌ No evidence of multi-video QA usage         |
> | InternVL3            | ❌ No usage examples with >1 video             |
> | Video-LLaMA 3        | ❌ No usage examples with >1 video             |
>
> Given these factors, we chose to benchmark **Gemini 2.5 Pro** first to test whether a model advertised as capable of multi-video understanding could perform spatial and temporal reasoning in cluttered real-world environments. The model’s poor performance on tasks like object localization validates the need for such targeted evaluation.
>
> That said, we fully agree that broader comparisons would strengthen the benchmark, and we are actively working on incorporating other models into the evaluation pipeline in future updates.
>
>
>
> ---
>
> ### Concern 2: Task overlap and unclear justification for five task splits
>
> We appreciate the reviewer’s feedback and agree that a clearer articulation of the five benchmark tasks would strengthen the paper. Each task is designed to isolate and evaluate a distinct capability of vision-language models (VLMs).
>
> To clarify the specific distinction between **Temporal Association VQA** and **Single-Cubicle-Multi-Temporal VQA**:
>
> * **Temporal Association VQA** is an *open-ended* task: the model is prompted to list all observed changes between two temporal episodes of the same cubicle. This setup removes any guidance or bias introduced by the question itself and serves as a *qualitative probe* of the model’s genuine understanding of the video content.
>
> * **Single-Cubicle-Multi-Temporal VQA**, in contrast, presents a *multiple-choice* question set based on annotated changes. It provides a structured evaluation of temporal understanding with *quantifiable accuracy metrics*.
>
> We emphasize that the **three core quantitative benchmarks** in the paper are:
>
> * **Static Association–Semantic Mapping VQA** – spatial understanding across one global video.
> * **Single-Cubicle-Multi-Temporal VQA** – fine-grained temporal reasoning.
> * **Multi-Cubicle-Multi-Temporal VQA** – joint spatial + temporal reasoning across scenes.
>
> Meanwhile, **Spatial Association VQA** and **Temporal Association VQA** serve primarily *diagnostic and explanatory* purposes. Below, we provide a summary of each task:
>
> | **Task**                                    | **Video Input**     | **Question Type** | **Primary Focus**                                                               |
> | ------------------------------------------- | ------------------- | ----------------- | ------------------------------------------------------------------------------- |
> | **Spatial Association VQA**                 | Single global video | Open-ended (JSON) | Tests cubicle enumeration and owner association in visually redundant layouts.  |
> | **Static Association–Semantic Mapping VQA** | Single global video | Multiple-choice   | Evaluates spatial reasoning without temporal complexity.                        |
> | **Temporal Association VQA**                | Local video pair    | Open-ended (JSON) | Assesses model’s ability to freely detect all object-level changes across time. |
> | **Single-Cubicle-Multi-Temporal VQA**       | Local video pair    | Multiple-choice   | Measures accuracy on predefined temporal changes within a single cubicle.       |
> | **Multi-Cubicle-Multi-Temporal VQA**        | Global video pair   | Multiple-choice   | Evaluates long-range spatio-temporal reasoning across multiple cubicles.        |
>
>
> ---
>
> ### Concern 3: Domain specificity and lack of dynamic elements
>
> Our benchmark focuses on **spatial-temporal associative reasoning**, which we define as the ability to link entities (e.g., objects and locations) across space and sparse time steps. Office cubicles provide an ideal testbed for this due to several factors:
>
> * **High visual redundancy**
>   Most cubicles look similar, requiring models to rely on sparse cues (e.g., name tags or cubicle codes) to resolve spatial references. This challenges the model’s ability to ground language in cluttered visual environments.
>
> * **Spatial + Temporal Association**
>   Consider the example of tracking *Veronica’s phone* as it is moved to another desk. To answer questions about this change, the model must:
>
>   1. Find Veronica’s cubicle and identify its boundaries (models often mistake objects near a cubicle as belonging inside it),
>   2. Locate the phone within her cubicle,
>   3. Recognize the same phone in a different cubicle in the next video,
>   4. Associate both cubicles with their respective owners.
>
>   This process requires the model to link object identity and ownership across space and time, making it a clear test of spatial-temporal association.
>
> * **Cluttered but structured scenes**
>   Offices contain many overlapping and occluded objects, providing a realistic and challenging environment for evaluating perception under ambiguity.
>
>
> We intentionally excluded **dynamics and human interaction** from this benchmark to isolate and evaluate spatial and temporal association capabilities. These aspects are outside the current scope and are better addressed by future benchmarks that focus on interaction-aware reasoning.
>
> Finally, office environments are highly relevant to real-world applications such as **surveillance**, **inventory monitoring**, and **workspace assistance**, where tracking object movement and ownership over time is a core requirement.
>
>
> ---
>
> ### Concern 4: Question generation via LLMs may introduce bias; only 20 questions per category were validated
>
> Due to the high labor cost of manual validation, we validated a random sample of **20 questions per category**. However, the **question generation process was tightly constrained and highly structured**, which reduces the likelihood of hallucinated or invalid outputs:
>
> * Changes were **logged in CSV files** with a fixed schema.
> * Prompts were built from **exact templates**, including specific object names, cubicle identifiers, and change types (e.g., count, location, state).
> * The output was **restricted to a multiple-choice format** (A–E), with randomized positions for the correct answer.
>
> This pipeline helps ensure consistency, correctness, and minimal LLM bias.
>
> That being said, we are currently in the process of **validating all questions** to guarantee full quality control in future releases.

---

### Official Review · Reviewer_y9Kc · 2025-07-03

**Rating:** 5
**Confidence:** 2

**Summary:**

This paper presents Office Hours, a new video benchmark dataset for evaluating Vision-Language Models (VLMs) on complex spatio-temporal reasoning tasks in real-world, cluttered office environments. Unlike existing benchmarks that focus on static and clean settings, Office Hours introduces:

1.Global: 6 robot-filmed walkthroughs across 23 cubicles over 5 temporal episodes.

2.Local: 215 short clips across 10 cubicles with 20 temporal episodes each.

**Dataset Code Accessibility:**

Yes

**Ethical Considerations:**

No, there are no or only very minor ethics concerns

**Final Justification:**

The discussion period did not change my score.

**Limitations Weaknesses:**

1. Evaluation is limited to Gemini 2.5 Pro. No comparison with open-source VLMs (e.g., GPT-4o, LLaVA, Qwen-VL), vision-only models, or traditional baselines.

**Strengths Contributions:**

1.Novel Real-World Benchmark: Fills a crucial gap in VLM evaluation by shifting from synthetic and static scenes to realistic, cluttered office environments. Scenes contain visually redundant cubicles, occlusion, ambiguous object ownership, and dynamic layouts—difficult conditions rarely covered in prior benchmarks.

2.Fine-Grained Task Decomposition: The five designed VQA tasks target specific weaknesses of VLMs: spatial ambiguity, temporal drift, and entity grounding. Each task is well-motivated and reflects practical robotic needs.

3.The benchmark rigorously tests both spatial and temporal object association at multiple scales.

---

> ### Author Rebuttal · Authors · 2025-07-31
>
> We appreciate the reviewer’s time and insights. Below, we provide a point-by-point rebuttal addressing the key concerns.
>
> ---
>
> ### Concern 1: Evaluation is limited to Gemini 2.5 Pro. No comparison with open-source VLMs.
>
> We acknowledge the limitation and appreciate the reviewer’s suggestion. Our decision to focus exclusively on **Gemini 2.5** Pro was aligned with the core objective of the Office Hours benchmark: to assess multi-video spatial and temporal reasoning in cluttered real-world scenes.
>
> To the best of our knowledge, **Gemini 2.5 Pro** is one of the only vision-language models (VLMs) that explicitly advertises support for multi-video input (1–10 videos per prompt) in its product documentation. This makes it particularly suited for our benchmark, which involves multiple cubicle walkthroughs and temporal video pairs.
>
> While models like **GPT-4o**, **LLaVA-Next**, **Video-LLaMA 3**, **Qwen2-VL**, and **InternVL3** are impressive, we found no documentation or examples indicating that they were designed for multi-video question answering.
>
> For completeness, we provide a summary below:
>
> | **Model**            | **Specifically Designed for Multi-Video QA?** |
> | -------------------- | --------------------------------------------- |
> | Gemini 2.5 Pro       | ✅ Claims support for 1–10 videos              |
> | GPT-4o               | ❌ Frame-based inference only                  |
> | LLaVA-Next           | ❌ No usage examples with >1 video             |
> | Qwen2-VL-7B-Instruct | ❌ No evidence of multi-video QA usage         |
> | InternVL3            | ❌ No usage examples with >1 video             |
> | Video-LLaMA 3        | ❌ No usage examples with >1 video             |
>
> Given these factors, we chose to benchmark **Gemini 2.5 Pro** first to test whether a model advertised as capable of multi-video understanding could perform spatial and temporal reasoning in cluttered real-world environments. The model’s poor performance on tasks like object localization validates the need for such targeted evaluation.
>
> That said, we fully agree that broader comparisons would strengthen the benchmark, and we are actively working on incorporating other models into the evaluation pipeline in future updates.

---

### Note · Authors · 2025-08-13

# Author Final Remarks

We thank the reviewers and the Area Chair for their thoughtful engagement. Across reviews, there was clear agreement that **Office Hours** offers a timely, realistic test of **spatio-temporal associative reasoning** in cluttered environments. Reviewers valued our **fine-grained task design**, **transparent collection and annotation protocols**, **open release**, and that experiments are **reproducible with modest compute (<\$300)**. Several comments also emphasized practical relevance to embodied AI (e.g., surveillance, inventory monitoring, workspace assistance).

**Key concerns and how we addressed them during discussion:**

1. **Model coverage.** We began with Gemini 2.5 Pro because, to our knowledge, it is the only VLM that explicitly advertises **multi-video input (1–10 videos)**, matching our multi-cubicle and temporal setup. We outlined plans to extend evaluations to additional models to strengthen comparisons.

2. **Task boundaries and clarity.** We clarified distinctions among the five tasks especially **Temporal Association** (open-ended listing of changes) versus **Single-Cubicle-Multi-Temporal** (multiple-choice over annotated changes). We emphasized three **core quantitative tracks** (Static Association–Semantic Mapping, Single-Cubicle-Multi-Temporal, Multi-Cubicle-Multi-Temporal), with the remaining tasks serving diagnostic/explanatory roles.

3. **Domain scope and modalities.** The office setting intentionally stresses association under redundancy, occlusion, and subtle state changes. We discussed transfer to other everyday domains (e.g., kitchens, living rooms) and future augmentation with **derived 3D cues** (e.g., point clouds from video) and, in later iterations, human dynamics while keeping the current RGB-centric evaluation aligned with how most VLMs operate today.

4. **Error analysis.**  We reported **per-category error breakdowns** (detection, counting, location, state) and linked multi-cubicle localization errors to upstream **boundary/ownership grounding** failures, providing a concrete root-cause view. We have also begun scaling comprehensive human validation.

**Takeaway.** The discussion increased alignment on the benchmark’s motivation and clarified design choices. We appreciate the reviewers who indicated positive or accepting stances, and we hope the community finds **Office Hours** a useful, extensible testbed for measuring and ultimately improving spatio-temporal associative reasoning in VLMs.

---

### Decision · Program_Chairs · 2025-09-18

**Decision:**

Reject

**Comment:**

The paper proposes a new benchmark to evaluate the performance of visual question answering by VLMs on spatio-temporal reasoning tasks in a dynamic, cluttered office environment. Some of the five proposed task types (spatial association, spatial reasoning without temporal changes, temporal association, and temporal changes in single- and multi-cubicle settings) are relatively complex, requiring temporal reasoning and disambiguation of similar environments. The dataset contains both global walkthrough data and local recording data of individual areas, and the associated VQA benchmark contains 1600+ multiple-choice and open-ended questions. Gemini 2.5 Pro was used as a baseline to demonstrate the utility of the benchmark.

Strengths:
- The dataset is drawn from a real-world environment, and as the authors point out, robotic environment benchmarks are often simplified. The data collection effort is substantial and well documented.
- The five VQA types capture a set of possible tasks that are both important to embodied reasoning and, in many cases, difficult for VLMs.
- Data collection, data, and code are clear and freely available, as required.

Weaknesses:
- Widely noted: Only one model is applied to the problem set, so the results may not be as generalizable as might be ideal. The authors note that relatively few models support video input; however, a similar experiment could have been performed using keyframes or the existing of video-language-models that have become available. The paper would be substantially strengthened by either the inclusion of these baselines or a detailed discussion of why these approaches are infeasible.
- The categorization choices for some of the tasks could be clearer; the subdivisions of tasks is not well defined in the paper itself. The table the authors provide in the rebuttal is informative, and could move to the paper itself.
- Questions were generated via LLM and only a small set of the questions (<10%) was manually validated. Although the templating of question generation decreases the likelihood of spurious questions, it also reduces their variety and complexity.
- Although robotics is used as a motivating case, and several examples involve robots, the authors describe the purpose of the benchmark as being primarily to evaluate VLMs, and omit the sensory complexity of robotics.

The paper is overall clear, the effort involved in the data collection was clearly substantial, and the benchmark for VLM capabilities is topical. However, the concerns about including more baselines (or a discussion of why existing video-LLMs or keyframe approaches are not suitable) would provide important context in both applicability of the benchmark to real-world tasks and appropriateness of the LLM-generated questions. I appreciate the authors’ detailed responses, but there’s a lot of information provided there that would strengthen a future version of the paper. I also note that the bar for NeurIPS this year is exceptionally high.